# Oilseed Cake Flour Composition, Functional Properties and Antioxidant Potential as Effects of Sieving and Species Differences

**DOI:** 10.3390/foods10112766

**Published:** 2021-11-11

**Authors:** Jan Bárta, Veronika Bártová, Markéta Jarošová, Josef Švajner, Pavel Smetana, Jaromír Kadlec, Vladimír Filip, Jan Kyselka, Markéta Berčíková, Zbyněk Zdráhal, Marie Bjelková, Marcin Kozak

**Affiliations:** 1Department of Plant Production, Faculty of Agriculture, University of South Bohemia, 370 05 České Budějovice, Czech Republic; barta@zf.jcu.cz (J.B.); jarosm08@zf.jcu.cz (M.J.); svajnj00@zf.jcu.cz (J.Š.); 2Department of Food Biotechnology and Agricultural Products Quality, Faculty of Agriculture, University of South Bohemia, 370 05 České Budějovice, Czech Republic; smetana@zf.jcu.cz (P.S.); kadlec@zf.jcu.cz (J.K.); 3Department of Dairy, Fat and Cosmetics, University of Chemistry and Technology, 166 28 Prague, Czech Republic; Vladimir.Filip@vscht.cz (V.F.); Jan.Kyselka@vscht.cz (J.K.); Marketa.Bercikova@vscht.cz (M.B.); 4Mendel Centre of Plant Genomics and Proteomics, Central European Institute of Technology, Masaryk University, 625 00 Brno, Czech Republic; zdrahal@sci.muni.cz; 5Department of Legumes and Technical Crops, Agritec Plant Research, Ltd., 787 01 Šumperk, Czech Republic; bjelkova@agritec.cz; 6Institute of Agroecology and Plant Production, Wrocław University of Environmental and Life Sciences, 50-363 Wrocław, Poland; marcin.kozak@upwr.edu.pl

**Keywords:** oilseed cake flour, protein, antioxidant activity, hemp, flax, milk thistle, poppy, sunflower, rapeseed

## Abstract

Oilseed cakes are produced as a by-product of oil pressing and are mostly used as feed. Their use for human consumption is due to the functional properties and benefits for human health. Herein, oilseed cake flours of eight species (flax, hemp, milk thistle, poppy, pumpkin, rapeseed, safflower, sunflower) were sieved into fractions above (A250) and below (B250) 250 µm. The chemical composition, SDS-PAGE profiles, colour, functional properties and antioxidant activities of these flours were evaluated. The B250 fractions were evaluated as being protein and ash rich, reaching crude protein and ash content ranging from 31.78% (milk thistle) to 57.47% (pumpkin) and from 5.0% (flax) to 11.19% (poppy), respectively. A high content of carbohydrates was found in the flours of hemp, milk thistle and safflower with a significant increase for the A250 fraction, with a subsequent relation to a high water holding capacity (WHC) for the A250 fraction (flax, poppy, pumpkin and sunflower). The A250 milk thistle flour was found to have the richest in polyphenols content (TPC) (40.89 mg GAE/g), with the highest antioxidant activity using an ABTS•+ assay (101.95 mg AAE/g). The A250 fraction for all the species exhibited lower lightness than the B250 fraction. The obtained results indicate that sieving oilseed flour with the aim to prepare flours with specific functional characteristics and composition is efficient only in combination with a particular species.

## 1. Introduction

Oilseed species are an important group of crop plants whose oil is edible and suitable for human consumption. Only a few species are significant in the total world trade out of about 40 different oil seeds [1]. The major oilseed producing areas are in the temperate zones. America and Europe together account for more than 60% of the world production. *Brassica* species are the second-largest oilseed crop after soybean in the world production, surpassing peanut, sunflower and cottonseed during the last two decades [2]. Minor oilseed crops that are cultivated in Central Europe include flax, hemp, milk thistle, poppy, pumpkin or safflower.

As a by-product of oil pressing, oilseed cakes are produced, which are mostly used as feed for farm animals because they contain significant amounts of protein and other valuable substances [3,4,5]. However, the increasing human population induces demand for more rational use of food processing by-products [5,6]. The compositions of oilseeds differ according to individual species or cultivars. In general, oilseeds and the produced oilseed cakes represent functional food ingredients, as they are rich in oleochemicals, phytochemicals with antioxidant activity, proteins, ash, fibre, carbohydrates, vitamins and minerals [6]. Oilseed cakes can serve as substrates for the production of bioactive compounds (proteins, dietary fibre, antioxidants) with beneficial characteristics for health that can be used in foods, cosmetics, textile and pharmaceutical industries. Moreover, they can also be used as substrates for the production of enzymes, antibiotics or mushrooms [5]. 

Predominantly, oilseed cakes and subsequently oilseed cake flour are valuable sources of gluten-free proteins [7] that are suitable alternatives for replacing animal or other plant sources of protein since they are easily digestible, non-toxic and nutritionally sufficient [7,8,9]. Cold press cakes of sunflower, flaxseed, pumpkin or hemp contain 19.9–44.9, 14.4–41.9, 29.4–53.9 and 23.5–33.6% crude proteins, respectively [5,7,9]. The pressed cakes could be transformed into flour, protein concentrates and protein isolates with protein contents of <65, 65–90 and >90%, respectively, as well as hydrolysates [6]. The obtained oilseed cake products with high protein content display high water-holding and fat absorption capacities, emulsifying and foaming activities and stability [7,10]. Protein concentrates and isolates are more suitable than oilseed cake flour for fortifying food products with plant proteins, but “wet” technology (including aqueous extraction, chemical use, drying) is needed to produce them, where these are technically and economically more demanding processes. The protein content of flour can be increased by separating the protein-rich fraction of fine particles from the fraction of coarser particles using dry methods. Sieving, which is one of the dry methods, can be used as a technically and economically inexpensive solution [11,12,13]. 

The use of oilseed cakes and flours for the fortification of various types of food products is already well described [9,14,15,16]. Knowledge of functional properties and composition is key to finding the optimal use, but this information is often very fragmented and the data presented are often limited to the evaluation of an individual species. For this reason, the aim of the present work was to assess the impact of sieving on the content and composition of nutritionally and biologically valuable components and selected functional properties in the oilseed cake flours of the eight selected oilseed crops, namely, flax (*Linum usitatissimum* L.), hemp (*Cannabis sativa* L.), milk thistle (*Silybum marianum* L.) Gaertn.), poppy (*Papaver somniferum* L.), pumpkin (*Cucurbita pepo* L. var. *oleifera*), rapeseed (*Brassica napus* L.), safflower (*Carthamus tinctorius* L.) and sunflower (*Helianthus annuus* L.). This work brings new information regarding the composition, functional properties and potential usability of various fraction sizes of the oilseed cake flours in relationship to the species. 

## 2. Materials and Methods

### 2.1. Industrial Processing of Oilseeds, Milling of Oilseed Cake and Preparation of Flour Fractions 

Industrial moistening and screw pressing of oilseeds, namely, flax (*Linum usitatis-simum* L.), hemp (*Cannabis sativa* L.), milk thistle (*Silybum marianum* (L.) Gaertn.), poppy (*Papaver somniferum* L.), rapeseed (*Brassica napus* L.), safflower (*Carthamus tinctorius* L.) and sunflower (*Helianthus annuus* L.) were done using direct saturated steam injection at 95 °C with a cold screw pressing operated at a temperature below 50 °C. The industrial processing of pumpkin seeds (*Cucurbita pepo* L. var. *oleifera*) consisted of milling, moistening at 70 °C, roasting (130 °C, 30 min) and cold pressing using a hydraulic press that was operated at a temperature below 50 °C. Cold oilseed cakes of eight oilseed species were vacuum packed and kindly donated by AGRO-EL Ltd. (Znojmo, Czech Republic) for further determination of the functional properties. 

The oilseed cakes were milled using the knife mill Grindomix GM200 (Retsch, Haan, Germany) under 10,000 rpm for 1 min and subsequently separated by sieving into size fractions (laboratory sieve system Retsch SATURN, Retsch, Germany). Finally, three size fractions of the oilseed cake flour were obtained: whole oilseed cake flour (WF) and oilseed cake flour fractions above 250 μm (A250) and below 250 μm (B250). The yields of the sieving fractions is given in Table 1. The contents of the dry matter, nitrogen components (crude protein), lipids (crude fat), carbohydrates, ash and moisture were determined in each of the flour fractions. Moreover, the obtained fractions were analysed for their SDS-PAGE protein profiles, water solubility, water and fat holding capacity, total phenolic content, antioxidant activity and colour profiles. All analyses were performed in triplicate.

### 2.2. Proximate Composition 

The ground and homogenised samples were analysed for water content using the gravimetric method (AOAC 925.10), crude protein content (N × 6.25) using the Dumas method, crude fat content using the Soxhlet method (AOAC 935.38) and ash content (AOAC 923.03) [17]. 

The moisture of the samples (2 g) was determined via drying them to a constant weight in a drying oven (Memmert oven U110, Memmert GmbH + Co. KG, Buechenbach, Germany), which was maintained at 105 °C for 3 h. 

The crude protein content was determined via the modified combustion method according to the Dumas method using the rapid N elementary analyser (Rapid N exceed, Elementar Analysen Systeme, Langenselbold, Germany). Each sample was analysed in triplicate and the crude protein (protein nitrogen) was calculated as nitrogen content multiplied by a factor of 6.25. 

The crude fat content was measured via the Soxhlet extraction method with petroleum ether using an automatic extraction system ANKOM XT 10 Extractor (ANKOM Technology, Macedon, NY, USA) according to the producer’s manual. Crude fat values were calculated from the weight difference before and after extraction. 

The ash content was measured as the total amount of inorganic residues remaining after the organic matter incineration by heating oilcake samples in a muffle furnace (muffle furnace LE 09/11, LAC s.r.o., Židlochovice, Czech Republic) at 550 °C for 5 h. 

The carbohydrates content was calculated as the remainder up to 100% of the sample dry matter (DM) after subtraction of the sum of all determined other component contents. 

### 2.3. Water Solubility and Water Holding Capacity 

Water solubility (WS) and water holding capacity (WHC) of the oilseed cake fractions were determined by mixing 300 mg of flour dry matter (DM) with 5 mL of deionised water. The tubes with the mixture were allowed to stand at room temperature for 30 min under continuous shaking. The samples were centrifuged (15 min, 3600× *g*, 20 °C) (ROTINA 420/420R, Hettich Instruments, Beverly, MA, USA). The supernatant was discarded and tubes with pellet were weighed, freeze-dried (−50 °C, <0.420 mbar, 48 h) (ALPHA 1–4 LSD, Martin Christ, Osterode am Harz, Germany) and weighed again. The solubility and water holding capacity were computed from the obtained weight data according to the following formulas (Equations (1)–(3)):WS (%) = 100 × (ms − mlp)/ms(1)
WHC (g/g) = (mwp − mlp)/ms(2)
WHCnon-soluble part (g/g) = (mwp − mlp)/mlp (3)
where WS is the water solubility of sample DM (%), ms is the weight of the sample (g), mlp is the weight of the freeze-dried pellet (g), WHC is the water-holding capacity of the samples (g of water per g DM), mwp is the weight of the wet pellet (g) and WHCnon-soluble part is the water-holding capacity of the DM of oilseed cake flour’s non-soluble part (g of water per g of non-soluble part DM). 

The values of WS and WHC after boiling were determined using the same analytical procedure as described for the unprocessed samples. The solubility (Equation (4)) and water holding capacity were computed from the obtained weight data: WHCnon-soluble part after boiling (g/g) = (mwpab − mlpab)/mlpab(4)
where WHCnon-soluble part after boiling is the water-holding capacity of the non-soluble part of flour after boiling (g of water per g of non-soluble part DM), mwpab is the weight of the wet pellet after boiling (g) and mlpab is the weight of the freeze-dried pellet after boiling (g).

### 2.4. Fat-Holding Capacity

The fat-holding capacity (FHC) was determined similarly to the WHC with some differences: 5 mL rapeseed oil was used instead of water. The supernatant (oil phase) was discarded after centrifugation and the tubes with the pellet were weighed. The FHC values were computed from the weight of the fatted pellet (Equation (5)): FHC (g/g) = (mfs − ms)/ms(5)
where FHC is fat-holding capacity (g of oil per g of DM), mfs is the weight of the fatted pellet (g) and ms is the weight of the sample (g).

### 2.5. Colour Analysis 

The colours of the oilseed cake flour samples were measured using a ColorEye^®^ XTH colourimeter (X-Rite, Grand Rapids, MI, USA), which is based on the CIE (Commission Internationale de l’Eclairrage) system and provides L*, a* and b* parameters (L*: lightness, 0%—black, 100%—white; a*: red—green; b*: yellow—blue). The analyses were performed in triplicate.

### 2.6. Total Phenolic Content 

The total phenolic content (TPC) was determined spectrophotometrically using Folin–Ciocalteu’s reagent after the previous extraction of phenolics using 80% ethanol with a solid-to-solvent ratio of 1:20. The extraction was performed over 24 h at room temperature and the mixture was then centrifuged (3600× *g*, 10 min), filtered and subsequently refrigerated at −20 °C until analysis. The extracts were analysed according to Lachman et al. [18] with some modifications. Briefly, 20 μL of extract and 1980 μL of distilled water were mixed with 100 μL of Folin-Ciocalteu’s phenol reagent, followed by the addition of 300 μL of 20% (*w/v*) sodium carbonate. The absorbance was read at λ = 750 nm (BioMate 5 spectrophotometer, Thermo Electron Corporation, Waltham, MA, USA) after 2 h. Gallic acid was used for calibration. All measurements were repeated three times and the results were expressed as milligrams of gallic acid equivalent (GAE) per gram of sample fresh matter (FM). 

### 2.7. Antioxidant Activity 

The oilseed fraction extracts for the determination of antioxidant activity were prepared as described for TPC (Section 2.6). Antioxidant activity was measured using the ABTS and DPPH methods according to Šulc et al. [19] with modifications. Ascorbic acid was used for the calibration. All measurements were repeated three times and the results were expressed as milligrams of ascorbic acid equivalent (AAE) per gram of sample FM. 

Radical scavenging activity with the ABTS•+ method: ABTS·+ radical was produced by dissolving 54.8 mg 2,2′-Azino-bis(3-ethylbenzothiazoline-6-sulfonic acid) diammonium salt (ABTS) and 1.0 g MnO_2_ in 20 mL of ultrapure water. The filtered solution (PTFE 0.25 μm) was diluted in 5 mM phosphate buffer to an absorbance of 0.800 ± 0.01 at λ = 734 nm. The absorbance of the reaction mixture was measured at λ = 734 nm 1 min after the addition of 100 μL of sample extract to 1 mL of the radical solution. 

Radical scavenging activity with the DPPH• method: 2,2-Diphenyl-1-picrylhydrazyl (DPPH) radical in an amount of 0.025 g was dissolved in 100 mL of methanol to obtain a stock solution. The reaction mixture was prepared from 975 µL of 10% (*v/v*) DPPH• stock solution and 25 µL of the analysed extract. The absorbance of the reaction mixture was measured at 515 nm after incubation at room temperature for 30 min.

### 2.8. Electrophoretic Analysis of Oilseed Cake Protein Profiles

Oilseed cake flour samples were extracted at 4 °C for 4 h using an SDS extraction buffer (0.065 M Tris-HCl, pH 6.8, 2% (*w/v*) SDS, 5% (*v/v*) 2-sulphanylethanol). Protein separation was carried out in duplicate using cooled dual vertical slab units SE 600 (Hoefer Inc., Holliston, MA, USA) with a discontinuous gel system (4% stacking and 12% resolving gel) in reduction conditions [20]. Protein detection was performed by using Coomassie Brilliant Blue R-250. 

### 2.9. Statistics 

The program Statistica 12 (StatSoft Power Solutions Inc., Palo Alto, CA, USA) was used for the data analysis. Data were subjected to analyses of variance using the two-way ANOVA method and the means were compared using the Fisher test. Differences between the variants were considered significant at *p* < 0.05 unless stated otherwise. 

## 3. Results and Discussion

### 3.1. Proximate Analysis of Oilseed Cake Flours 

The approximate compositions of all variants that were derived from the oil cake flours are given in Table 2. The contents of crude proteins, crude fat, ash, carbohydrates and moisture were evaluated using two-way ANOVA (Fisher’s LSD test). Significant differences were found for most of the variants. The crude protein content in the whole oilseed cake flour (WF) ranged widely from 14.21% (safflower) to 59.12% (pumpkin) of the FW. Except for pumpkin, the sieving process increased the crude protein content. The size fractions with finer particles (B250) reached a higher proportion of crude protein and these differences were statistically significant in most cases. The crude protein level of the fine fractions ranged from 31.78% (milk thistle) to 57.47% (pumpkin). The highest differences between the A250 and B250 fractions were observed for species hemp, safflower, milk thistle and sunflower. Insignificant differences between the size fractions were found for poppy and pumpkin. The significant increase in crude protein content in several species (hemp, milk thistle, safflower and sunflower) in the B250 fraction resulted from the different structures of their seeds, including the pericarp. In agreement with our results, Zając et al. [9] and Lan et al. [21] reported an average crude protein content ranging from 24 to 28% for non-dehulled hemp seeds and Shen et al. [18] determined 32.7 and 41.8% of crude protein content for non-dehulled and dehulled hempseed flour, respectively. Similarly, from the data of Murru and Calvo [13], sunflower seed hulls made up to 30% of the seed weight and have poor nutritional properties due to their minimal content of protein and lipids. Sunflower hull contains only 3% protein and 61% fibre, while fully dehulled sunflower meal contained up to 63% protein and only 9% fibre, as presented by Murru, Calvo [13]. 

The effect of size fractionation on crude protein content was limited for other species (flax, poppy, pumpkin or rapeseed), which can be given in connection with the absence of pericarp. This conclusion is well documented regarding carbohydrates content and their distribution between the size fractions for individual species. Carbohydrates contents, including fibre, were significantly lower in the group of flax, poppy, pumpkin and rapeseed and differences in carbohydrates content between the A250 and B250 fractions were more pronounced in the group of oilseed species with a pericarp. The lowest content of carbohydrates (approximately 14% for all size fractions) and the highest content of protein (ranged from 57.7 to 60.4%) was observed for pumpkin. The usual level of crude protein content in pumpkin seeds range from 20 to 40% [22,23,24]. Nikolić et al. [23] reported a protein level of about 30–40% in hull-less pumpkin seeds with a natural content of oil and milled cake flour obtained from hull-less pumpkin seeds consisting of concentrated proteins (55–65%). The results of Nikolić et al. [25] and Kreft et al. [26] are in agreement with our results. Such a product with a high protein content could be usable in food applications for enhancing food products’ nutritive value [23,25]. 

In contrast, we found a high content of carbohydrates in the flour of hemp, milk thistle and safflower, with a significant increase for the A250 fraction. Differences in carbohydrates content were minimal and statistically insignificant for the flours of poppy and pumpkin. Dietary fibres are considered to be one of the major ingredients that are used to develop products with a functional purpose [9,27] and oilseed cake products with increasing content of this component can be added to new health benefit products. 

Residual crude fat content in the WF fraction varied from 4.84% (milk thistle) to 16.16% (flax) of the FW. The amount of residual fat increased with decreasing flour particle size. The observed differences for the WF and A250 fractions were statistically significant, except for sunflower. The second exception was safflower flour, with the highest content of residual fat in the WP fraction and the lowest level in the B250 fraction. The highest differences in fat content between the A250 and B250 fractions were determined for flax and rapeseed. A high content of residual fat in the B250 fraction appears to be problematic due to the application of this protein-rich source. On the one hand, the increased fat content of flax flour can be used as a substitute for animal fat sources (e.g., butter), thereby increasing the health, environmental and economic benefits of such products. On the other hand, it is necessary to take into account the instability of flax fat components and the possible link to flour technological quality. Bochkarev et al. [28] proposed a 25% proportion of fat in oilseed cakes as the maximum level, as the higher fat content in the flour leads to faster oxidation, and at the same time, the mixing possibility with other components becomes worse. The instability of the fat component is also typical for hemp seed oil with a high content of polyunsaturated fatty acids (PUFA) [29]. However, the PUFA have several health benefits for the consumers and the potential for enhancing the n3/n6 acids ratio in food products [9]. 

The tested oilseed cake flours also differed regarding the ash content. The species with a pericarp (hemp, milk thistle, safflower, sunflower) exhibited higher differences in ash content between individual size fractions. The lowest contents were determined in the A250 size fraction, and the highest level was found in the B250 fraction. This phenomenon was also observed for poppy and rapeseed flours; however, the differences between the fractions were minimal in comparison with the seed-coated species. Ash content in pumpkin flour was specific in comparison with other species. The lowest content of ash (8.94% FM) was examined in the B250 fraction, while 9.14 and 9.42% FM were found in the WF and A250 fractions, respectively. The ash content in the protein-rich fraction (B250) ranged among the species from 5.0% (flax) to 11.19% FM (poppy). For wheat flour [30], the technical quality of high-ash flour is lower because it is characterised by a darker colour and greater activity of proteolytic and amylolytic enzymes. However, in the case of oilseed cake flours, the results are unclear. The determined ash content in the analysed flours was in accordance with the results of other authors. Bochkarev et al. [28] reported ash content in ranges of 4.8–5.0 and 5.0–5.6% FM for flax and milk thistle flour, respectively. Similarly, the high content of ash in poppy flour was in agreement with Yilmaz and Emir [31], who specified that the high ash level in poppy was caused by the K, P, Mg and Ca contents. This represents an additional potential in increasing the health and nutritional benefits of oilseed cake flours and their products.

### 3.2. Electrophoretic Characteristics of Oilseed Cake Flour Proteins 

The SDS-PAGE profiles of oilseed cakes within individual size fractions are shown in Figure 1. Electrophoretic bands covering the apparent 5–100 kDa range were found for all analysed samples, with visible protein pattern variability between individual species and size fractions. The increase in protein pattern intensities between the A250 and B250 fractions were evident, especially for hemp, milk thistle and safflower. These observations correlated with the data that were related to the change in crude protein content between the A250 and B250 size fractions that were obtained for individual species (Table 2). The flax SDS-PAGE profile was typical for this species, creating four main areas of protein bands that occurred approximately between 50 and 10 kDa. The detected protein zones were therefore identical to those reported by other authors [14,32]. The protein band of about 50 kDa that was found in all size fractions of flaxseed flour could be described as one of five subunits of globulin linin. The 11–12S polypeptide subunits of this protein have molecular weights of 14.4, 24.6, 30.0, 35.3 and 50.9 kDa [14] and bands with these molecular weights were found in the flax seed SDS-PAGE profiles, as can be seen in Figure 1. Subunit alpha shows more significant variability in the B250 fraction, where three protein bands of the alpha subunit were visible. 

The protein profiles that were detected within other species could be similarly characterised into the globulin and albumin protein groups with various manifestations. Globular edestin, which is the main storage protein of hemp as the hexamer protein, was visible in all hemp variants in the form of acidic and basic subunits that were detectable in the zones of 34 and 20 kDa (designated as the “subunits alpha and beta” zones). This was in agreement with Kotecka-Majchrzak et al. [14]. The difference between the hemp A250 and B250 fractions was clear in the SDS-PAGE profiles, particularly in the beta subunits. Similar differences between the A250 and B250 fractions were visible for milk thistle. Li et al. [33] determined that milk thistle albumin and globulin (molecular mass range of 16–112 kDa) were the dominant protein fractions. This information is largely in agreement with our SDS-PAGE profile. The protein bands of milk thistle were found in the range from ~10 to 110 kDa and the bands predominantly cumulated in the zone of subunits alpha and beta. These major protein bands probably belong to albumin and globulin fractions, as reported by Li et al. [33]. 

Similar SDS-PAGE profiles to those for milk thistle were detected for safflower, sunflower and partly similar for rapeseed. The protein profiles ranged from ~10 to 100 kDa, where a high number of individual protein bands cumulated in the area of subunits alpha and beta. The safflower profiles also showed a different intensity of protein bands in these major areas, with a higher intensity for the B250 fraction. Zones of cruciferin (area of subunits alpha and beta) and napin (area of 2S albumin) were found in the rapeseed SDS-PAGE profiles. Cruciferin, oleosin and napin are the main storage proteins of rapeseed, with high potential for use in functional foods and food substitutes production [14,34]. The sunflower SDS-PAGE profiles did not show differences between the flour size fractions. The 11S helianthinin and 2S albumin, which were reported previously for this species [35], are clearly visible in the zones of subunits alpha, beta (helianthinin) and 2S albumin (albumin fraction). 

The poppy and pumpkin SDS-PAGE profiles were similar to the hemp and flax samples. A somewhat higher intensity of protein bands was found in the B250 fraction. Bučko et al. [36] determined 12S globulin cucurbitin (six subunits of 54 kDa) and 2S albumin (12.5 kDa) to be major proteins of pumpkin, together forming 59% of the total protein. However, we also found significant protein bands in the zone of subunits alpha and beta, with values of molecular weight of 30–36 and 15–19 kDa, respectively. These data correspond with the conclusions of Hara et al. [37], who presented pumpkin globulin separation into two subunits (α and β), corresponding to weights of 63 and 56 kDa, respectively. Via the reduction of disulphide bonds, the two subunits were each separated into two polypeptide chains with molecular weights of around 36 and 22 kDa.

### 3.3. Functional Characteristics of Oilseed Cake Flours 

Table 3 shows the functional properties of oilseed cake flours in a relation to their species origins and particle size distributions after flour sieving. The ANOVA analysis showed the significant effect of the species and flour size fractionation on the observed parameters: water solubility (WS), water-holding capacity (WHC), water holding capacity after boiling (WHCafter boiling) and fat-holding capacity (FHC). The values of WS were increased for all evaluated variants with increasing fineness of the flour. The highest WS values were thus achieved for the B250 fraction for all evaluated species. In contrast, the A250 fraction had a decrease in solubility compared with the B250 fraction and for milk thistle, rapeseed, sunflower and the WF fraction (with statistical significance). The highest value of WS (33.12%) was found for the B250 rapeseed flour, with the other two fractions of this species also achieving a high solubility when compared to the other species. The B250 fraction was characterised as a protein-rich fraction that had a rapeseed protein content of 33.52% FM, which may explain the high solubility. The rapeseed storage proteins consisted of approximately 50% cruciferin (known as 11S globulin) and 20–40% of the albumin napin fraction (2S albumin), which is very soluble in water [38,39]. The presence of these proteins was also confirmed in our samples using SDS-PAGE analysis (Figure 1). The WS value was thus species-specific, given by the level and structure of the globulin and albumin protein fraction and the content of minerals and soluble carbohydrates.

Similarly, the dominant albumin protein fraction was described in milk thistle [30] and the increase in solubility of the B250 fraction in this species can be linked to this fact. The solubility of flax was the second highest and achieved 26.68%. In contrast, the B250 fraction of hemp and safflower flours showed the lowest solubility, achieving 18.98 and 17.30%, respectively. The albumin fraction constituted about 25% of hempseed storage protein, while the globulin fraction (edestin) represented up to 80% [40] and the albumin fraction exhibited significantly higher solubility than the globulin one [41]. Malomo et al. [42] observed that the hemp protein fraction solubility gradually increased with increasing pH. This suggests a reduction in protein solubility during protein isoelectric precipitation via protein complexation. The protein-rich fraction that was obtained by sieving with a more native state of protein fraction was more appropriate from this perspective. 

The water-holding capacity values ranged within the evaluated set from 1.44 g/g FW (safflower, A250 fraction) to 4.90 g/g FW (flax, A250 fraction). Low WHC values (<2.00 g/g) were found for hemp, pumpkin and safflower. The second group with WHC values between 2.19 and 3.3 g/g FM consisted of milk thistle, rapeseed and sunflower. The highest WHC was observed for all size fractions of flax flour, ranging from 3.17 g/g FM (B250 fraction) to 4.15 g/g FM (A250 fraction). This was probably related to the content of flaxseed gum, which is a constituent of dietary fibre present in the flaxseed hull [43]. This flaxseed component has a function of a food hydrocolloid and significantly increases the WHC of foods, which is important for both the yield and texture of related products [44]. The obtained values of the WHC that correspond with the carbohydrates content are presented in Table 2. The WF and A250 fractions contained significantly higher amounts of carbohydrates in comparison with the protein-rich B250 fraction. Similarly, the lowest WHC was obtained for the B250 fraction. The WF and A250 fractions with high WHC can therefore be used as a component of products where water binding is an important functional indicator. Water holding affects the texture, juiciness, taste of food formulations and, in particular, the shelf-life of bakery products [45]. Boiling oilseed flours increased their WHC, except for poppy flour; however, the trend of WHC that was observed for the unboiled variants also manifested itself in the boiled variants. The values of WHC below 2 g/g FM of the sample were observed for boiled flours of hemp, pumpkin and safflower. Medium WHC values were found for milk thistle, poppy, rapeseed and sunflower. Very high values of WHC were observed for all the variants of flaxseed flour, which were again the highest for the A250 fraction and lowest for the B250 fraction. These results also confirmed the finding that partial hydrolysis (enzymatic or heat treatment) can improve WHC that was reported previously [46]. 

Very similar values were found within the evaluated variants for fat-holding capacity. The lowest FHC values were obtained for pumpkin flour variants with values ranging between 0.65 g/g FM (WF and A250 fractions) and 0.71 g/g FM (B250 fraction). Higher FHC values were obtained for flax, ranging from 1.09 g/g FM (A250 fraction) to 0.91 g/g FM (B250 fraction). As for the WHC, the presence of flax gums seemed to be related to the fat-binding ability. The values of flax FHC corresponded with the carbohydrates content. Comparable high values of FHC were also recorded for poppy and sunflower, while the lowest FHC values were recorded for WF in the majority of the species, except for flax and rapeseed. However, it is necessary to notice that the differences in the FHC between the size fractions were minimal and, in many cases, not statistically significant (hemp, pumpkin, rapeseed, sunflower). 

### 3.4. Total Polyphenols Content and Antioxidant Activities of Oilseed Cake Flours 

The total polyphenols content and values of antioxidant activities (ABTS radical cation-based and DPPH radical) are presented in Table 4. The results indicated significantly different contents of phenolic components in the flours of the analysed oilseed species but also highly different distributions between the size fractions, which were dependent on the individual species. Species with low TPCs (≤3 mg GAE/g FM) were hemp, poppy and pumpkin; species with medium TPCs (approximately between 3 and 18 mg GAE/g FM) were flax, rapeseed, safflower and sunflower. Similarly, size fractionation divided the oilseed species into two groups according to the TPC. The first one was composed of flax, hemp and milk thistle, where the highest TPC was found for the A250 fraction (the differences were not statistically significant for hemp samples), while the second one consisted of poppy, pumpkin, rapeseed, safflower and sunflower. These results indicated different distributions of the TPC or other antioxidant components in seeds of individual oilseed species. 

The highest TPC was found in the flour of milk thistle, which ranged between 17.68 mg GAE/g FM (B250 fraction) and 40.89 mg GAE/g FM (A250 fraction). The key component of milk thistle polyphenols are bioactive flavonolignans, which are referred to as silymarin or the silymarin complex [47,48]. Silymarin is accumulated mostly in the seed coat [49]. Similarly, flax seed polyphenols, namely, the lignans, occur in the seed coat compartments [50]. This explains the detection of the highest TPC in the A250 flour fraction in both species. On the other hand, the binding of polyphenolic components on protein fraction during its isolation and extraction process was described for sunflower [51] and rapeseed [52]. Moreover, the majority of the sunflower polyphenols is contained in the nucleus, while only a small amount is found in the packaging structures [53]. These facts explain the higher TPC in the fraction with a lower grain size.

A similar trend as for TPC was also found for antioxidant activity of the analysed flour samples within the individual evaluated variants (Table 4). Using the ABTS•+, the lowest levels of antioxidant activity were recorded for poppy samples, ranging from 3.02 mg AAE/g FM (WF) to 3.66 mg AAE/g FM (A250). Low values of antioxidant activity were observed also for pumpkin and hemp flours. The highest antioxidant activity that was found using the ABTS•+ was observed for milk thistle flour. High antioxidant activities were examined within all size fractions of this species; however, the A250 fraction had the highest activity, which reached 101.95 mg AAE/g flour. The antioxidant activity of milk thistle seeds was previously given in correlation with bioactive flavonolignans (silymarin complex) [48,54]. Moreover, our data suggested the accumulation of antioxidants in the surface structure of milk thistle seeds. These findings represent a significant shift in the possibility of using milk thistle flour, and especially the A250 fraction. The mentioned fraction, despite its significantly lower protein content, is usable as a source of substances with antioxidant and protective potential. Increasing of antioxidant potential of food products by adding whole seed flour of milk thistle was previously described [55,56]. Using milk thistle A250 flour can enhance the antioxidant potential of milk thistle flour even more. Its application in composite flours with flax, hemp or pumpkin could increase their antioxidant activity; it can also be used in products that need to be protected against oxidation (e.g., meat products). The milk thistle A250 fraction with the above-mentioned characteristics can also be used for the production of functional foods for specific human diets, as these products can also offer chemoprotective and hepatoprotective effects [51]. The significantly higher antioxidant activity of the A250 flax fraction (17.45 mg AAE/g flour) in comparison with other flax flour size fractions also gives this fraction interesting potential for exploitation. Flax and hemp flours are often prone to oxidation due to their higher content of unsaturated fatty acids [57,58]. This issue can be solved using flax flour defatting or adding components with a higher antioxidant potential, e.g., A250 fraction flax flour. 

Protein-rich B250 fractions with high antioxidant activities of 20.51, 33.82 and 31.03 mg AAE/g flour were found for rapeseed, safflower and sunflower, respectively. The antioxidant activities of these fractions fit with the observed TPC values. 

The data of antioxidant activities that were obtained using the DPPH• radical method approximately followed the trend of antioxidant activity that was determined with the ABTS radical; however, a decrease in measured values was observed. The lower sensitivity of the antioxidant assay using the DPPH radical is typical and was published previously [59,60]. The lower sensitivity of the DPPH lipid peroxide radical scavenging assay is also obvious from the lower level of the statistically significant differences between size fractions, as well as between species, e.g., statistically significant differences between all evaluated size fractions were found only for rapeseed flour. The highest antioxidant activity using the DPPH• radical was found for the A250 fraction of milk thistle, reaching 11.37 mg AAE/g, and no significant difference was found between the A250 and B250 fractions. Similarly, high antioxidant activity against DPPH• was found for rapeseed and sunflower B250 fractions, reaching 11.05 and 15.49 mg AAE/g flour, respectively. In agreement with the results of ABTS•+, DPPH• antioxidant activities were also at very low levels for poppy and pumpkin (under 1 mg AAE/g flour). Antioxidant activities on medium values using the DPPH• radical (between 1 and 4 mg AAE/g) were found for hemp, flax and safflower.

Interesting in this case may be the comparison of the content of polyphenols and the antioxidant activity of the oilseeds with other known sources of antioxidants. Wine, for example, is considered to be an important donor of polyphenols with a high value of antioxidant capacity. The TPCs and antioxidant activity that was obtained by using DPPH• in grape seeds were 546 mg/g DM and 1.0 AAE mg/mL, respectively [61]. In contrast, low TPC and antioxidant activity (DPPH•) were observed in potato tubers, reaching averages of only 3.8 mg GAE/g DM and 0.25 mg AAE/g DM. Here, however, it is necessary to take the high consumption into account [62].

### 3.5. Colour Characteristics of Oilseed Cake Flours 

The concentration of biologically active components alongside the antioxidant activities (e.g., phenolic substances or some pigments) is often connected with the colour characteristics. However, there has been some disagreement about the relationship between the total phenolic content and antioxidant activity. Liu et al. [63] found that wheat grain colour does not appear to be a factor that is related to the antioxidant parameters. On the other side, colour is a key factor that influences customer selection [64] and the usability of alternative flours in the food industry. The colour parameters (CIE L*, a* and b* coordinates) of oilseed flour samples are given in Figure 2. The species and fractions of oilseed flours had significant effects on the colour scores i.e., the L* (lightness/darkness), a* (redness/greenness) and b* (yellowness/blueness). The lightness/darkness observed in the oilseed flour samples differed from the darkest with an L* value of 51.9 (hemp flour) to the lightest with an L* value of 63.0 (milk thistle). The L* parameters were significantly affected by the size fractionation in all the evaluated species. The A250 fraction for all the species exhibited lower lightness in comparison with the B250 fraction. The L* value of the A250 and B250 fractions ranged between 41.85 (poppy) and 56.36 (flax) and between 51.88 (poppy) and 68.0 (rapeseed), respectively. The oilseed flour fractionation can thus contribute to colour optimisation when a darker colour of products may ultimately lower the consumer’s acceptability [65].

The greenish colour (−a*) was found only for whole seed flour of rapeseed and a very low a* value was found for sunflower. The size fractionation also significantly affected the a* and b* colour parameters. The greenish tones (−a*) did not appear in any sample of either A250 or B250 size fraction. However, a very low a* value was found for rapeseed and sunflower in both the size fractions. The highest value of reddish tones in the A250 fraction was found for flax flour (a = 6.06) and for poppy in the B250 fraction (a = 5.40). The fractionation had a significant effect on yellowness, with the exception of sunflower. In general, with the increasing fineness of the flour, the lightness and yellowness increased. This dependence was very distinct in rapeseed, with the yellow values a* being 8.46 and 26.81 for the A250 and B250 fractions, respectively. Within the B250 fraction, rapeseed flour also showed the most yellow colour compared with the other species. 

The lowest value of the b* parameter was found in the B250 fraction of sunflower. Flour of this species showed yellow and red intensities within the B250 group, as well as in the A250 fraction, with minimal differences between the size fractions. 

The obtained results indicated that sieving oilseed flour with the aim to improve the functional and visual properties was efficient only in combination with a particular species. The significant effect of supplemented oilseed flour on colour scores was previously described, e.g., flaxseed flour addition significantly decreases the lightness and increased the redness [64]; green and yellow tones are typical for pumpkin seed flour [25]; defatted sunflower flour is significantly darker with higher redness and blueness values [66] and hemp flour addition significantly decreases the lightness, redness and yellowness of bread crust [65]. 

## 4. Conclusions 

This study produced complex information about the composition, functional and antioxidant properties of oilseed cakes of eight oilseed species (flax, hemp, milk thistle, poppy, pumpkin, rapeseed, safflower and sunflower) in three size flour fractions: whole oilseed cake flour, flour size fraction above 250 µm (A250) and below 250 µm (B250). Such complex data has not been published before. 

In conclusion, the content characteristics were significantly influenced both by the oilseed species and by the size fraction of the flour samples. The highest content of proteins, as the key component from a nutritional and functional perspective, was found in pumpkin flour with an equal level in all three fractions. With the exception of pumpkin and poppy, the B250 fraction could be described as being the richest in protein, where sieving is a way of increasing the protein content. Antioxidant activity was affected by both the species and the fraction. Flax, hemp and milk thistle species exhibited significantly higher polyphenols content and antioxidant activity in the A250 fraction, while for safflower and sunflower, these properties were the highest in the B250 fraction. In all the tested species, the fractionation significantly influenced the lightness of the flour samples. The lightness was improved by the sieving: the finer the flour, the lighter the colour. Yet even regarding the colour, species specificities emerged.

The obtained data confirmed the potential of mechanical sieving for some of the tested oilseed species to produce flours with specific characteristics, e.g., flour rich in protein or flour rich in antioxidants that are usable in various applications. 

## Figures and Tables

**Figure 1 foods-10-02766-f001:**
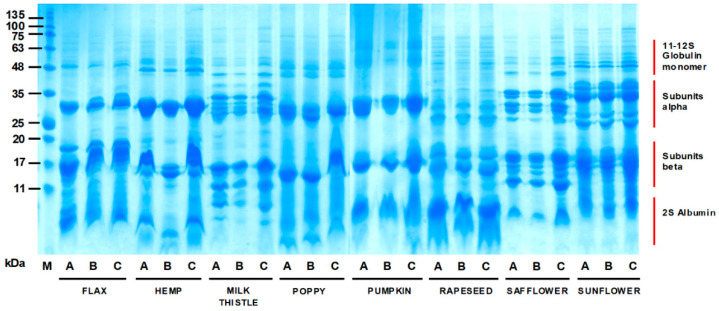
SDS-PAGE protein profiles of the oilseed cake flours under reducing conditions: M—molecular weight marker in kDa; A—whole oil cake flour; B—flour fraction above the 250 µm size; C—flour fraction below the 250 µm size.

**Figure 2 foods-10-02766-f002:**
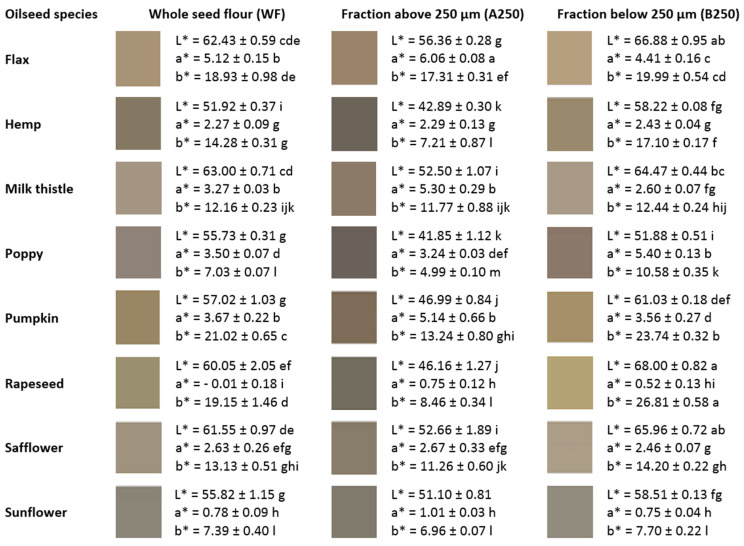
Colour characteristics of oilseed cake flours: CIE parameters: L, lightness; a, red—green; b, yellow—blue; different letters indicate a statistically significant difference at *p* < 0.05 (Fisher’s LSD test).

**Table 1 foods-10-02766-t001:** Yields of the sieving fractions.

Oil Cake	Yield (%)	
	A250	B250
Flax	51.95	45.15
Hemp	61.99	35.20
Milk thistle	50.01	49.06
Poppy	41.41	56.50
Pumpkin	38.04	59.79
Rapeseed	49.57	48.00
Safflower	68.45	30.64
Sunflower	56.53	41.51

**Table 2 foods-10-02766-t002:** Composition analysis of oil cake flours from eight selected oil crops.

Oilseed Cake	Crude Protein (%)	Crude Fat (%)	Carbohydrates (%)	Ash (%)	Moisture (%)
Flax					
WF	28.46 ± 0.44 hi	16.16 ± 0.11 b	41.74 ± 0.37 j	4.73 ± 0.03 o	8.92 ± 0.10 b
A250	24.81 ± 0.60 j	13.03 ± 0.26 d	47.97 ± 0.29 gh	4.80 ± 0.01 o	9.38 ± 0.10 a
B250	33.58 ± 0.47 f	19.56 ± 0.10 a	33.65 ± 0.44 m	5.00 ± 0.04 n	8.21 ± 0.15 cde
Hemp					
WF	28.25 ± 0.42 hi	8.61 ± 0.24 i	50.23 ± 0.66 f	5.34 ± 0.02 m	7.56 ± 0.33 fg
A250	18.83 ± 2.42 l	6.30 ± 0.14 m	62.19 ± 2.31 d	4.22 ± 0.08 p	8.46 ± 0.09 c
B250	45.69 ± 0.44 c	10.24 ± 0.11 g	27.84 ± 0.41 o	7.65 ± 0.04 g	8.58 ± 0.12 bc
Milk thistle					
WF	22.56 ± 0.36 k	4.84 ± 0.23 o	63.73 ± 0.23 d	5.78 ± 0.03 jk	3.09 ± 0.66 l
A250	10.75 ± 1.20 n	2.27 ± 0.44 p	78.17 ± 1.36 a	3.61 ± 0.03 q	5.20 ± 0.05 k
B250	31.78 ± 1.61 g	5.46 ± 0.21 n	49.49 ± 1.74 fg	8.11 ± 0.04 f	5.16 ± 0.50 k
Poppy					
WF	34.75 ± 0.44 ef	10.49 ± 0.13 g	36.03 ± 0.43 kl	10.90 ± 0.01 b	7.84 ± 0.15 ef
A250	34.66 ± 0.36 ef	9.75 ± 0.05 h	37.32 ± 0.23 k	10.94 ± 0.09 b	7.33 ± 0.09 gh
B250	35.42 ± 0.49 e	11.25 ± 0.23 f	34.62 ± 0.67 lm	11.19 ± 0.02 a	7.52 ± 0.03 fg
Pumpkin					
WF	59.12 ± 0.47 a	12.36 ± 0.26 e	14.39 ± 0.67 p	9.14 ± 0.06 d	4.99 ± 0.14 k
A250	60.38 ± 0.31 a	8.95 ± 0.06 i	14.67 ± 0.39 p	9.42 ± 0.11 c	6.58 ± 0.18 i
B250	57.47 ± 0.46 b	13.65 ± 0.05 c	14.06 ± 0.65 p	8.94 ± 0.03 e	5.88 ± 0.31 j
Rapeseed					
WF	29.84 ± 1.50 h	12.52 ± 0.19 e	43.84 ± 1.67 i	5.73 ± 0.12 k	8.08 ± 0.14 de
A250	24.96 ± 0.54 j	8.82 ± 0.42 i	52.30 ± 0.76 e	5.49 ± 0.08 l	8.43 ± 0.09 cd
B250	33.52 ± 0.18 f	15.89 ± 0.13 b	37.42 ± 0.35 k	5.90 ± 0.02 ij	7.27 ± 0.28 gh
Safflower					
WF	14.21 ± 1.20 m	8.00 ± 0.16 j	69.30 ± 0.92 c	2.51 ± 0.31 r	5.98 ± 0.03 j
A250	8.38 ± 1.97 o	7.17 ± 0.15 k	75.19 ± 2.10 b	2.07 ± 0.03 s	7.20 ± 0.03 gh
B250	34.29 ± 1.06 ef	6.75 ± 0.09 l	46.72 ± 0.86 h	5.11 ± 0.04 n	13 ± 0.12 h
Sunflower					
WF	34.41 ± 0.51 ef	12.47 ± 0.07 e	40.52 ± 0.77 j	5.99 ± 0.01 i	6.60 ± 0.34 i
A250	27.87 ± 1.60 i	11.12 ± 0.64 f	48.38 ± 1.44 gh	5.25 ± 0.08 m	7.37 ± 0.05 gh
B250	41.66 ± 0.22 d	12.20 ± 0.01 e	31.49 ± 0.27 n	7.42 ± 0.04 h	7.24 ± 0.06 gh

WF—whole oilseed cake flour; A250—oilseed cake flour fraction above 250 µm; B250—oilseed cake flour fraction below 250 µm. Different letters in the columns indicate a statistically significant difference at *p* < 0.05 (Fisher’s LSD test).

**Table 3 foods-10-02766-t003:** Selected functional properties of oil cake flours.

Oil Cake	WS (%)	WHC (g/g of Flour)	WHC after Boiling (g/g of Flour)	FHC (g/g of Flour)
Flax				
WF	21.84 ± 2.49 d	4.15 ± 0.24 b	8.12 ± 0.15 a	0.97 ± 0.01 c
A250	22.26 ± 0.92 cd	4.90 ± 0.03 a	8.32 ± 0.06 a	1.09 ± 0.04 a
B250	26.68 ± 1.26 b	3.17 ± 0.29 cd	7.42 ± 0.37 b	0.91 ± 0.01 d
Hemp				
WF	15.36 ± 0.52 jkl	1.67 ± 0.04 kl	1.91 ± 0.15 hij	0.77 ± 0.03 jk
A250	13.08 ± 0.23 lm	1.61 ± 0.06 lmn	1.70 ± 0.01 jkl	0.82 ± 0.01 hi
B250	18.98 ± 1.66 efgh	1.83 ± 0.08 k	1.98 ± 0.05 hi	0.83 ± 0.01 hi
Milk thistle				
WF	15.95 ± 0.68 ijk	2.73 ± 0.09 f	2.99 ± 0.16 d	0.83 ± 0.01 hi
A250	12.89 ± 0.31 m	2.19 ± 0.09 j	2.58 ± 0.18 ef	0.85 ± 0.00 gh
B250	20.60 ± 1.58 def	3.33 ± 0.09 c	3.18 ± 0.20 cd	0.90 ± 0.02 de
Poppy				
WF	21.13 ± 0.71 de	2.67 ± 0.09 fg	2.35 ± 0.02 fg	0.91 ± 0.01 d
A250	19.34 ± 0.41 efgh	3.07 ± 0.08 de	2.55 ± 0.09 ef	0.99 ± 0.03 bc
B250	20.87 ± 0.47 def	2.55 ± 0.17 ghi	2.16 ± 0.14 gh	1.01 ± 0.01 b
Pumpkin				
WF	19.91 ± 1.37 defg	1.45 ± 0.06 mn	1.65 ± 0.06 kl	0.65 ± 0.01 m
A250	18.11 ± 4.96 ghi	1.60 ± 0.13 lmn	1.84 ± 0.06 ijk	0.65 ± 0.01 m
B250	19.10 ± 0.13 efgh	1.46 ± 0.06 mn	1.55 ± 0.06 l	0.71 ± 0.03 l
Rapeseed				
WF	25.08 ± 1.59 b	2.42 ± 0.19 hi	3.10 ± 0.15 cd	0.90 ± 0.01 de
A250	20.62 ± 0.64 def	2.37 ± 0.05 ij	3.33 ± 0.34 c	0.86 ± 0.07 efgh
B250	33.12 ± 0.35 a	2.42 ± 0.02 hi	2.40 ± 0.28 fg	0.89 ± 0.02 def
Safflower				
WF	13.56 ± 0.54 klm	1.56 ± 0.04 lmn	1.95 ± 0.07 hij	0.74 ± 0.01 kl
A250	12.31 ± 1.78 m	1.44 ± 0.03 n	1.88 ± 0.08 ijk	0.79 ± 0.01 ij
B250	17.30 ± 0.58 hij	1.63 ± 0.04 lm	1.89 ± 0.06 ijk	0.85 ± 0.02 fgh
Sunflower				
WF	22.31 ± 1.67 cd	2.66 ± 0.07 fg	2.67 ± 0.07 e	0.86 ± 0.00 efgh
A250	18.54 ± 0.28 fgh	2.96 ± 0.05 e	2.99 ± 0.22 d	1.03 ± 0.03 b
B250	24.63 ± 0.96 bc	2.58 ± 0.04 fgh	2.49 ± 0.04 ef	0.88 ± 0.03 defg

A250—oilseed cake flour fraction above 250 µm; B250—oilseed cake flour fraction below 250 µm; WF—whole oilseed cake flour; WS—water solubility; WHC—water-holding capacity; FHC—fat-holding capacity. Different letters in columns indicate a statistically significant difference at *p* < 0.05 (Fisher’s LSD test).

**Table 4 foods-10-02766-t004:** Total phenolics contents and antioxidant activities of oilseed cake flours.

Oilseed Cake	Total Phenolics Content (mg GAE/g)	Antioxidant Activity ABTS•+ (mg AAE/g)	Antioxidant Activity DPPH• (mg AAE/g)
Flax			
WF	6.13 ± 0.59 hi	13.50 ± 1.15 f	2.35 ± 0.03 f
A250	7.42 ± 0.41 fg	17.45 ± 0.34 e	2.55 ± 0.03 f
B250	3.59 ± 0.28 j	8.16 ± 0.25 gij	1.84 ± 0.01 g
Hemp			
WF	2.72 ± 0.13 jk	7.70 ± 0.51 ijk	1.66 ± 0.04 g
A250	2.99 ± 0.33 jk	9.36 ± 0.53 gi	1.96 ± 0.03 g
B250	2.61 ± 0.37 jkl	8.34 ± 0.43 gij	1.25 ± 0.06 h
Milk thistle			
WF	30.44 ± 0.42 b	86.42 ± 6.43 b	11.36 ± 0.56 b
A250	40.89 ± 1.60 a	101.95 ± 4.14 a	11.37 ± 0.43 b
B250	17.68 ± 1.20 c	55.71 ± 1.42 c	9.63 ± 0.65 c
Poppy			
WF	2.84 ± 0.36 jk	3.02 ± 0.07 l	0.89 ± 0.01 i
A250	2.39 ± 0.12 klm	3.66 ± 0.21 l	0.90 ± 0.00 I
B250	3.02 ± 0.34 jk	3.61 ± 0.24 l	0.87 ± 0.03 i
Pumpkin			
WF	1.45 ± 0.11 m	4.34 ± 0.19 kl	0.92 ± 0.02 hi
A250	1.41 ± 0.12 m	5.20 ± 0.13 jkl	0.90 ± 0.05 i
B250	1.52 ± 0.18 lm	6.19 ± 0.31 ijkl	0.90 ± 0.04 i
Rapeseed			
WF	8.47 ± 0.56 f	17.42 ± 0.47 e	9.33 ± 0.08 c
A250	6.97 ± 0.23 gh	11.56 ± 1.22 fg	7.74 ± 0.13 d
B250	10.51 ± 1.03 e	20.51 ± 1.66 e	11.05 ± 0.09 b
Safflower			
WF	6.79 ± 0.47 gh	19.95 ± 0.15 e	3.07 ± 0.02 e
A250	5.49 ± 0.29 i	17.59 ± 0.59 e	2.90 ± 0.13 e
B250	11.99 ± 1.15 d	33.82 ± 1.13 d	3.07 ± 0.04 e
Sunflower			
WF	17.21 ± 0.89 c	17.67 ± 6.51 e	15.37 ± 0.11 a
A250	12.93 ± 0.40 d	18.85 ± 2.47 e	15.29 ± 0.17 a
B250	17.65 ± 1.19 c	31.03 ± 1.26 d	15.49 ± 0.09 a

A250—oilseed cake flour fraction above 250 µm; B250—oilseed cake flour fraction below 250 µm; GAE-gallic acid equivalent; AAE–ascorbic acid equivalent. Different letters in columns indicate a statistically significant difference at *p* < 0.05 (Fisher’s LSD test).

## Data Availability

The data that support the findings of this study are available from the corresponding author upon reasonable request.

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
