# Peer review of "Oilseed Cake Flour Composition, Functional Properties and Antioxidant Potential as Effects of Sieving and Species Differences"

_foods, 2021, doi:10.3390/foods10112766_

Round 1
Reviewer 1 Report
The manuscript titled “Oilseed cake flour composition, functional properties and antioxidant potential as the effect of sieving and species differences” is related to (i) oilcake valorisation from 8 plants, (ii) further oilcakes processing including milling and sieving, (iii) chemical and technological parameters assessment of obtained flour fractions. Usage of oilcakes in combination with sieving in order to obtain fractions with different chemical compositions (e.g. protein rich) is not novel, thus paper has a lack of novelty. On the other hand, paper could be a source for a lot of data in one place. Anyhow, there are major issues which should be addressed.
Major:
Beside important chemical characterization of obtained sieving fractions, their yield is equally as important particularly if a manuscript is oriented towards potential industrialization of the process. Thus, authors should insert a part related to fraction yields for all plant matrices.
Section 2.1. Authors utilized different types of seeds, thus pressing parameters could differ between plants and it is important to provide some of information about pressing process. Pre-treatment conditions, type of press, working conditions, pressing capacity etc.
Minor:
Page 2 Lines 87-89 – Please add sieves producer
Section 2.2. Please provide more detailed information (sample weights, producers of equipment where missing, number of replicates where missing etc.) and references for proximate composition analysis.
Page 3 Line 116 – please provide a full term for DM as it is mentioned the first time
Page 3 Lines 118-119 – please provide a producer of a centrifuge and freeze drier
Page 4 Line 167 – I assume that used wavelength was 750 nm, 734 nm is usually utilized for ABTS test.
Page 5 Line 220 – “seeds” instead of “fruits” would be a better term
Page 8 Line 338 – from ~10 kDa to 110 kDa
Page 8 Line 342 from ~10 kDa to 100 kDa
Section 3.4. Please compare antioxidant power (this might be difficult due to different units) and TPC of obtained fractions with some fruits/vegetable which are well known for their antioxidant potential. It would give a general idea about the results presented in a study.
Page 13 Line 500 – with the higher antioxidant
Figure 2 title is a copy of Figure 1, should be corrected.
Page 15 Line 580 – Flax, hemp etc are not considered as fruits, should be corrected in the entire manuscript.
Author Response
Dear reviewer,
Thank you for your comments on the manuscript. Please find attached a list of comments and their settlement.

Reviewer 2 Report
I have some comments and suggestions for authors.
All are listed below with an appropriate Line number(s) from text in order to facilitate tracking:
Line 19: Suggest to replace "by their" with " "due to".
Lines 28-29: Precise here what antioxidant assays were applied.
Line 40: Brassica is plant genus. So, it should be given in italic style. Please correct.
Line 92: Delete s in phenolics. It is surplus here since you have term total in front of word.
Line 163: Put g in Italic here.
Line 167: Split numerical value from = with space.
Line 180: Please correct chemical formula for MnO2 to be correctly written.
Line 209: "proteins" in plural?
Lines 224 and 228: It should be "Murru and Calvo". Correct.
Line 232: It should be "carbohydrates content". Correct.
Line 285: I think it should be in not on here? Check/correct.
Line 297: It should be "... in accoradance with results of ... ". Correct.
Line 300: It should be "Yilmaz and Emir". Correct.
Line 300: It should be "is caused by" instead of "is given". In my opinion it is much better term here in this context.
Line 443: Actually, ATBS is radical cation not just radical like DPPH. It should be correct here and after in the text. Apply to all.
Line 474: Missing . at the end of sentence. Please correct.
Line 478: observed not examined here.
Line 480: The same as previous.
Line 540: The wrong label for Figure. Please check and provide adequate one.
Line 575: "proteins" in plural?
Lines 609-778: I think that list of references is not fully in accordance with MDPI rules. For instance, journal names should be given in Italic style. Please check Instruction for authors once again.
Author Response
Dear reviewer,
Thank you for your comments on the publication. Please find attached a list of comments and their settlement.

Reviewer 3 Report
This manuscript delas with the assessement of bioactive content in oilseed press cake of eight oil species. The topic of this manuscript may be considered as important for the valorization of the oil seed cakes in this case.
The most important results concerns the color characterization.
Nevertheless, it is dificult, when several species are studied to avoid the presentation as "a listing".
The most shortcoming is the perspective of use of such cakes. This is really limited in this manuscript.
Please see some examples in Foods
https://www.mdpi.com/search?q=bread+fortification&journal=foods
Author Response

(The authors gave the same response as above.)

Round 2
Reviewer 1 Report
The authors improved the manuscript. Prior to acceptance, authors should move table 1 in 3. Section and present it as 3.1 in which they should comment on the yields of fractions. Also, proximate analysis should be renumbered to 3.2 and so on. In table 1 dots should be used as decimal separators.